# Measuring Steerability in Large Language Models

**Trenton Chang**[1*]   **Jenna Wiens**[1]   **Tobias Schnabel**[2]   **Adith Swaminathan**[3]
[1]University of Michigan   [2]Microsoft Research   [3]Netflix
{ctrenton, wiensj}@umich.edu

## Abstract

Large language models (LLMs) are powerful instruction followers. However, many open-ended generation tasks have a large solution space that needs to be narrowed down to fit user needs. LLMs that are *steerable* towards such needs are critical to safe LLM systems that behave consistently with user expectations and goals. Despite continued improvement in LLM instruction-following, such gains may not necessarily translate to steerability. This disconnect motivates a principled framework for measuring steerability. Thus, we propose a goal-oriented, quantitative definition of steerability. Our definition informs the design of an empirical steerability probe, where we leverage text rewriting tasks to measure steerability of LLMs. We demonstrate that recent LLMs are not steerable. We attribute this lack of steerability to *side-effects*: correlations between requested goals and non-requested LLM movement. Thus, despite advances in LLM instruction following, there remains significant room for improving LLM steerability.

## 1 Introduction

Large language models (LLMs) have empirically demonstrated powerful instruction-following abilities in a variety of domains (Achiam et al., 2023; Dubey et al., 2024). However, many real-world tasks such as text editing or creative writing are open-ended: the desired set of generations depends on user expectations or needs. Users performing such tasks require flexible and controllable LLMs that can be steered to match their diverse needs.

Although LLMs continue to improve at instruction following, better instruction following may not imply better steerability. Instruction following datasets in current evaluations focus on coarse, impersonal changes, often opting for instructions about simplifying/summarizing text (Shu et al., 2024) or syntactical/linguistic constraints (Zhou et al., 2023). Moreover, even when prompted to produce changes in tone (*e.g.*, increased sadness, politeness), an LLM could still default to a congenial, "American" tone due to dataset biases during pre-training or reinforcement learning from human feedback (RLHF). Thus, current benchmarks could report good instruction following performance even if an LLM is not steerable towards producing texts with variation in tone.

Steerability is critical for designing safe LLMs that align with a variety of human values and priorities (Sorensen et al., 2024). This is because safety in one context does not imply safety in another: blanket refusals for legal or medical advice can mitigate harms from misinformation, but could also impede model usability. A context-aware view of LLM safety should prioritize LLMs that can easily navigate different desiderata. Tasks where steerable LLMs are key to safety may include clinical note generation (Abacha et al., 2023) and model-assisted psychotherapy (Sharma et al., 2024).

In this work, we propose a definition of LLM steerability, casting user goals as vectors (Section 2). As an example of an open-ended generation task, we focus on measuring steerability in single-turn text-rewriting. Figure 1 (left) illustrates our proposed definition of steerability via a text simplification request. The user's request and LLM's attempt to satisfy the request can be interpreted as vectors in *goal-space*, with dimensions comprised of textual aspects (Figure 1, right). Deviations between the two vectors in distance and direction map to steerability failures: incorrect distances

---

*Work done as an intern at Microsoft Research.

Figure 1: Text rewriting (left) can be modeled as movement in a vector space of goals (right). We propose two steerability metrics (sensitivity and directionality) that quantify distance and direction-based closeness between user requests (green) and LLM movement (orange) in goal-space. Source text from *Meditation XVII, Devotions Upon Emergent Occasions*, John Donne (1624).

may signify insufficient or exaggerated changes to text with respect to the user's request, while differences in direction may arise from changes in unanticipated aspects of text.

Using our steerability definition, we design an empirical steerability probe and apply it to multiple LLM families (Section 3). Our probe demonstrates that larger models are not necessarily steerable. We attribute this lack of steerability to *side-effects*: requesting one goal may yield shifts in goals not mentioned in the prompt. Our analysis highlights steerability as an independently important axis of LLM performance, and lays the groundwork for well-principled steerability evaluation.

## 2 DEFINING STEERABILITY

In this section, we define steerability, and disaggregate it into *distance* and *direction*-based components (Section 2.1). Then, we design a steerability probe based on our definition (Section 2.2).

### 2.1 STEERABILITY METRICS

Since we focus on text rewriting, we first define a space of textual goals to standardize measurement of textual change. Let $\mathcal{G}$ be a set of "goals" (*e.g.*, happiness level, reading difficulty), where each string maps to some point in **goal-space** $\mathcal{Z} \subseteq \mathbb{R}^{|\mathcal{G}|}$; *i.e.*, a string's **goal-space mapping** is a real-valued vector. Thus, text changes can be modeled as vectors in goal-space. Concretely, let $\mathbf{z}^* \in \mathcal{Z}$ be a user goal, and let $\mathbf{z}_0 \in \mathcal{Z}$ be the goal-space mapping of the source text, yielding user request vector $\mathbf{z}^* - \mathbf{z}_0$. Let $\hat{\mathbf{z}}$ be the goal-space mapping of the LLM output, such that $\hat{\mathbf{z}} - \mathbf{z}_0$ describes an LLM's movement in goal-space. Our steerabilty metrics compare user requests ($\mathbf{z}^* - \mathbf{z}_0$) to the LLM's goal-space movement ($\hat{\mathbf{z}} - \mathbf{z}_0$). Thus, we propose the following definition:

$$\text{Steerability}(\text{LLM}, \mathcal{D}) \triangleq \mathop{\mathbb{E}}_{\mathbf{z}_0, \mathbf{z}^* \sim \mathcal{D}} \left[ \frac{(\mathbf{z}^* - \mathbf{z}_0)^\top (\hat{\mathbf{z}} - \mathbf{z}_0)}{\|\mathbf{z}^* - \mathbf{z}_0\|_2^2} \right]. \tag{1}$$

We assume $\hat{\mathbf{z}}$ is a deterministic function of $\mathbf{z}_0$ and $\mathbf{z}^*$ (*e.g.*, a prompt based on $\mathbf{z}^*$ plus source text corresponding to $\mathbf{z}_0$). Eq. 1 defines steerability as the magnitude of the vector projection of the LLM's goal-space movement ($\hat{\mathbf{z}} - \mathbf{z}_0$)) onto the user request vector ($\mathbf{z}^* - \mathbf{z}_0$), normalized by the magnitude of the request vector ($\|\mathbf{z}^* - \mathbf{z}_0\|_2$). In other words, steerability quantifies the LLM's progress in goal-space as a proportion of the user's request vector. Thus, a value of 1 is necessary for steerability (*i.e.*, the projection of $\hat{\mathbf{z}} - \mathbf{z}_0$ onto $\mathbf{z}^* - \mathbf{z}_0$ is simply $\mathbf{z}^* - \mathbf{z}_0$). This definition also suggests that a steerable LLM should output text that "aligns" with the user's request in goal-space, *and* transforms the source text by an appropriate amount. We can decompose our steerability metric in terms of these two desiderata:

$$\text{Sensitivity}(\text{LLM}, \mathcal{D}) \triangleq \mathop{\mathbb{E}}_{\mathbf{z}_0, \mathbf{z}^* \sim \mathcal{D}} \left[ \frac{\|\hat{\mathbf{z}} - \mathbf{z}_0\|_2}{\|\mathbf{z}^* - \mathbf{z}_0\|_2} \right] \tag{2}$$

$$\text{Directionality}(\text{LLM}, \mathcal{D}) \triangleq \mathop{\mathbb{E}}_{\mathbf{z}_0, \mathbf{z}^* \sim \mathcal{D}} \left[ \frac{(\mathbf{z}^* - \mathbf{z}_0)^\top (\hat{\mathbf{z}} - \mathbf{z}_0)}{\|\mathbf{z}^* - \mathbf{z}_0\|_2 \|\hat{\mathbf{z}} - \mathbf{z}_0\|_2} \right]. \tag{3}$$

Note that steerability (for a single example) is equal to the product of sensitivity and directionality. Thus, for a model to be steerable by definition, it needs to be close to 1 for both metrics. Intuitively,

if $\|\mathbf{z}^* - \mathbf{z}_0\|_2 < \|\hat{\mathbf{z}} - \mathbf{z}_0\|_2$, the LLM is may be "overshooting": if the angle between $\|\mathbf{z}^* - \mathbf{z}_0\|_2$ and $\|\hat{\mathbf{z}} - \mathbf{z}_0\|_2$ is small, the LLM may be exaggerating desired changes to text. Note that $\|\mathbf{z}^* - \mathbf{z}_0\|_2 > \|\hat{\mathbf{z}} - \mathbf{z}_0\|_2$ implies the opposite. A large angle between $\mathbf{z}^* - \mathbf{z}_0$ and $\hat{\mathbf{z}} - \mathbf{z}_0$ may signal "side-effects" in requesting a goal, such that asking for changes in certain goals induces changes in unrelated goals. We summarize our steerability metrics in Figure 1.

Note that we define steerability in terms of $\mathbf{z}_0$ (Eq. 1): steerability for source texts in a small subset of goal-space does not imply general steerability for all source texts. An input-agnostic definition of steerability (*e.g.*, $\mathbb{E}_{\mathbf{z}^*}[\|\mathbf{z}^* - \hat{\mathbf{z}}\|_2]$) could fail to capture steerability across goal-space. Our two-metric view of steerability (Eq. 2 & 3) also allows us to disaggregate different steerability failures with respect to LLM inputs.

## 2.2 IMPLEMENTING A STEERABILITY PROBE

We design a steerability probe following our framework. We begin with a seed set of source texts, from which we define a set of goal dimensions and measurement methods. We then sample a diverse dataset $\mathcal{D}$ of starting goals ($\mathbf{z}_0$) and ending goals ($\mathbf{z}^*$). This probe is held fixed and applied across LLMs to compare steerability across models.

**Sampling diverse texts.** We concatenate subsets of four English-language datasets to collect a diverse seed set of source texts: CNN/Dailymail ($N = 1220$, See et al. (2017)), BookSum ($N = 2133$, Kryściński et al. (2021)), Reddit TIFU ($N = 2991$, Kim et al. (2018)), and SCROLLS (SummScreenFD only, $N = 338$, Shaham et al. (2022)). These datasets encompass expository (CNN/Dailymail, SCROLLS), narrative (Reddit TIFU), and creative writing (BookSum), as well as formal (CNN/Dailymail, SCROLLS, BookSum) and colloquial (Reddit TIFU, BookSum) English.

**Defining dimensions of goal-space.** We aim to sample a diverse set of texts in goal-space. To map texts to goal-space, we choose a set of goal dimensions, and use existing models and metrics that map strings to scalars. This yields a vector-based representation of an arbitrary text. Leveraging existing models and metrics to compute goal-space mappings means that goal-space mappings are computed identically for all LLMs, and do not require knowledge of the underlying LLM parameters.

As goal-dimensions, we select reading difficulty (Flesch-Kincaid grade level; Kincaid et al. (1975)), text diversity (Median Lexical Text Diversity; Jarvis & Hashimoto (2021)), text length (word count), six different tones corresponding to different emotions (via a sentiment classifier; Hartmann (2022)), and politeness (via a weighted sum of politeness strategies; Danescu-Niculescu-Mizil et al. (2013)). We also measure (but do not manipulate) six aspects of toxicity as measured by the Detoxify model (Hanu & Unitary team, 2020). Scores for the six emotions, toxicity aspects, and politeness were computed by sentence, then averaged. This yields 10 requestable goals and 6 toxicity-related goals. We use these models and metrics to map all texts to goal-space.

To produce a diverse sample of texts, we normalize all goal dimension via linear scaling such that the middle 95% of values observed in our seed set of texts maps to $[0, 1]$. We compute uniform sampling weights for the 10 non-toxicity related goal dimensions via classifier-based density ratio estimation (Bickel et al., 2007). Further dataset processing details are in Appendix A.1.

**Sampling user goal vectors.** User goal vectors are 10-dimensional vectors in $[0, 1]^{10}$, with each dimension corresponding to our 10 requestable goals. To create goal vectors, we randomly choose 3 goal dimensions for each text and sample target goal *values* $z_i^*$ ($i$th component of $\mathbf{z}^*$) as follows:

$$z_i^* := z_{0,i} + \Delta_i; \quad \Delta_i \sim \mathcal{U}([\max\{-z_{0,i}, -0.7\}, -0.1] \cup [0.1, \min\{0.7, 1 - z_{0,i}\}]) \qquad (4)$$

where $z_{0,i}$ is the $i$th component of $\mathbf{z}_0$, *i.e.*, we sample target goal to lie between 0.1 and 0.7 in absolute distance to the original value ($z_{0,i}$). We set $\Delta_i = 0$ in goal dimensions not chosen for modificiation. The $\min$ and $\max$ clip $z_i^*$ to the range $[0, 1]$. We sample 40 goal vectors per text in a sample of 50 source texts for a total of 2000 (text, goal) tuples.

To convert $\mathbf{z}^*$ into natural language prompts, we use a template-based prompt that asks for "slightly more (less)", "more (less)", or "much more (less)" of each goal dimension (examples in Appendix A.2). We also conduct a sensitivity analysis of prompting with a more granular prompt (specifying changes on a 1-10 scale; Appendix B.2).

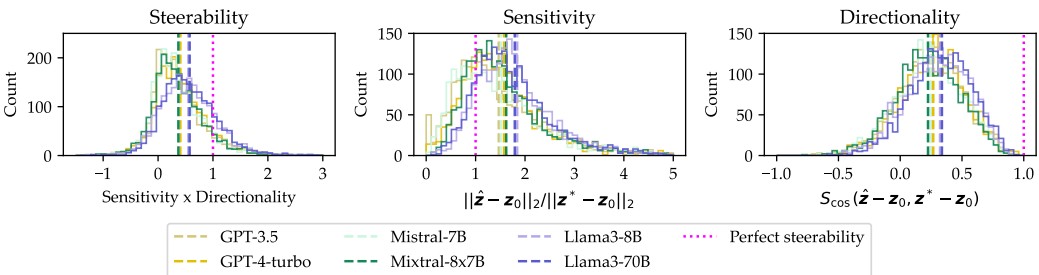

Figure 2: Steerability (left), sensitivity (middle) and directionality (right) histograms for GPT (yellow), Mistral (green), and Llama (blue) model families. Vertical dashed lines indicates means for each model. Pink dotted line indicates optimal values. $S_{\cos}$ : cosine similarity.

Our empirical steerability probe allows us to test a variety of hypotheses on whether LLM characteristics or design choices affect steerability. As a representative example, consider the impact of model size on steerability. Some evaluations find that larger models are better instruction-followers (*e.g.*, Chatbot Arena ELO, MT-Bench score (Zheng et al., 2023)), suggesting that larger models could be more responsive to user requests.[1] Concurrent work in prompt engineering finds that larger models are less sensitive to prompt paraphrases (Schnabel & Neville, 2024), implying the opposite. Our probe provides a means to investigate whether larger models are more or less steerable.

## 3  ARE LARGER MODELS MORE STEERABLE?

**Experimental setup.**  To evaluate whether larger models are more steerable, we compare smaller versions of models with larger versions in three model families: GPT-3.5 vs. -4 turbo (unknown changes;  Achiam et al. (2023)), Mistral/Mixtral (non-mixture model vs. mixture-of-experts; Jiang et al. (2023; 2024)), and Llama3 (8B vs. 70B;  Dubey et al. (2024)). We assume that GPT-4 is larger than GPT-3.5.[2]  The sampling temperature is set to 0. As a quality check, we manually examine a subset of generated LLM outputs to ensure that the request was handled correctly (Appendix A.4).

### 3.1  BIGGER MODELS ARE NOT ALWAYS MORE STEERABLE

Our results indicate a negligible relationship between model size and steerability metrics, suggesting that larger models are not always more steerable. Figure 2 shows histograms of steerability metrics for GPT (3.5 vs. 4, in yellow), Llama 3 (8B vs. 70B, in blue), and Mistral (7B vs. 8x7B/mixture of experts, in green).[3]  The high overlap between histograms across all models suggests that the models evaluated exhibit similar levels of steerability, including models of different size/generation within the same family. The standard deviation (SD) of all metrics is also large (*e.g.*, for GPT-4, SD(steerability) = 0.47, SD(sensitivity) = 0.92, SD(directionality) = 0.27). All models evaluated have steerability lower than 1, which is attributable to low directionality: the mean directionality for models evaluated ranges from 0.22 (Mixtral) to 0.33 (Llama3-70B). However, mean sensitivity exceeds 1 for all models, suggesting that models potentially "overshoot" in goal-space.

Differences in steerability are also larger between families of LLMs than within LLM families. GPT-3.5 and 4 have mean steerability values of 0.37 and 0.41 (difference: 0.04), respectively, while Llama-8B and -70B obtain values of 0.55 and 0.58 (difference: 0.03), respectively. Yet the directional awareness gap between the Llama and GPT model families is 0.14, which is $3.5\times$ to $4.7\times$ larger than the within-family directional awareness gap. Similar trends hold for sensitivity and directionality. Thus, factors other than model size may be responsible for the variation (albeit small) in steerability across models in our steerability probe.

---

[1]See `https://lmarena.ai/?leaderboard`, under "Full Leaderboard."

[2]We refer to GPT-4 turbo as "GPT-4." While some sources have reported parameter counts of order $\sim 10^{10}$ to $10^{11}$ for GPT-3.5 and of order $\sim 10^{11}$ to $10^{14}$ for GPT-4, we were unable to independently corroborate these figures from publicly-available, peer-reviewed sources.

[3]We provide a version of Figure 2 disaggregated by model family in Appendix B.1.

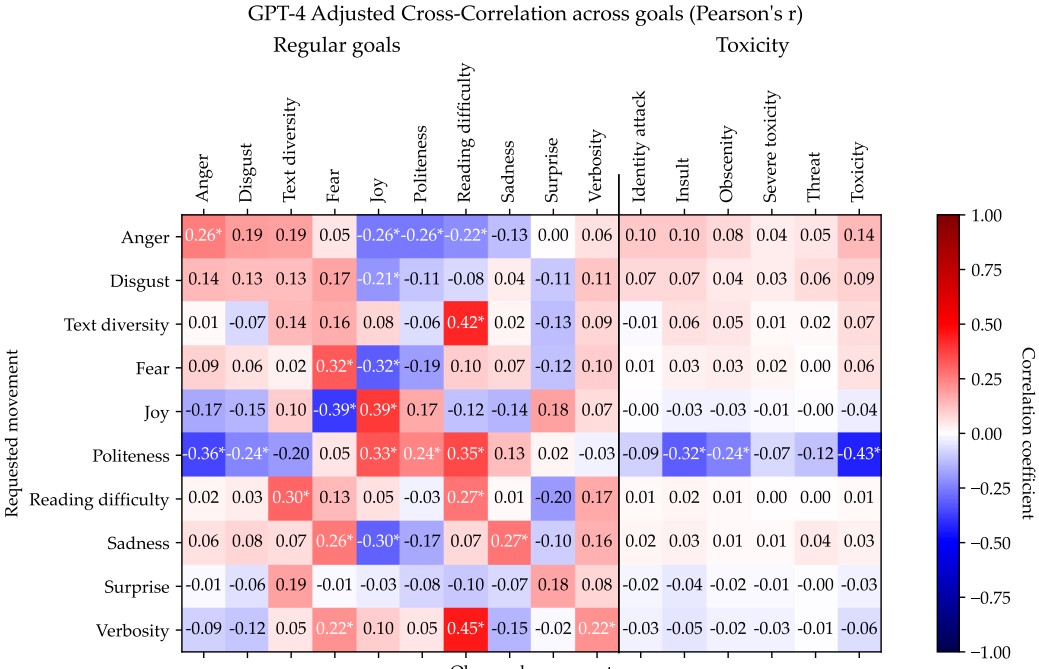

Figure 3: Adjusted cross-correlation ($\rho$) between requested movement (vertical axis) and observed movement (horizontal axes) across goal dimensions, divided by non toxicity-related (left) and toxicity-related (right) dimensions. Negative (positive) correlations in blue (red); darker shades denote larger-magnitude correlations. "*" denotes statistical significance ($\alpha = 0.05$ with Bonferroni correction; $n = 160$).

Ultimately, our probe demonstrates that larger models are not necessarily more steerable. In particular, the small gaps in steerability metrics across different model generations within the same family suggest updates to LLM development (*e.g.*, model pre-training/fine-tuning and RLHF techniques) may not map to improvements in steerability.

## 3.2 LLMs MOVE IN EXTRANEOUS DIRECTIONS IN GOAL-SPACE

**Requesting goals comes with side-effects.** We hypothesize that, since our prompts only specify a subset of possible goals, the prompt is implicitly *underspecified*. Thus, we may observe extraneous correlations between requested movement in goal-space and movement in other goals: a *side-effect* of requesting a particular goal. Such behavior could potentially yield LLMs with high sensitivity, yet poor directionality, as previously observed.

To investigate side-effects, we compute the adjusted Pearson cross-correlation between requested goals and measured goals. Since the goal-space mapping of a source text potentially leaks information about the target goal $\mathbf{z}^*$ (*e.g.*, a text in an extremal region of goal-space has less "range of movement" than a less-extreme text), an unadjusted cross-correlation metric could overestimate the effect of requested goals on goal-space movement. We correct for this by subtracting the cross-correlation between goals observed under an uninformative prompt, which we further discuss in Appendix A.3.

Figure 3 shows the adjusted Pearson cross-correlation ($\rho$) between requested vs. observed goal-space movement for GPT-4. Off-diagonal elements significantly larger than zero are potential side-effects. While some correlations are expected (*e.g.*, increasing verbosity is associated with greater reading difficulty, $\rho = 0.45$), others may not always be desirable. For example, increasing sadness is associated with greater fear ($\rho = 0.26$), while greater politeness is associated with greater reading difficulty ($\rho = 0.35$). Similar correlations persist in other models (Appendix B).

Note that seven of ten diagonal elements of the adjusted cross-correlation matrix are significantly larger than zero. A larger-than zero adjusted correlation signifies that, compared to an uninformative prompting strategy, requests to change a particular goal are more positively correlated with move-

ment in the same goal. In other words, LLMs appear to successfully follow most instructions when requested, consistent with empirical evidence of their strong instruction-following abilities.

**Are side-effects features or bugs?** Our results suggest that the non-steerability of LLMs is not due to poor instruction-following or sensitivity to model inputs, but a previously-undocumented phenomenon: side-effects. These changes could be appropriate: for example, inverse correlations between politeness and disgust ($\rho = -0.24$, Figure 3) in the tone of a text are subjectively suitable.

Yet other side effects do not reflect semantically inherent relationships. Fear and sadness, while negative in sentiment, are not equivalent. Similarly, politer text is not always harder to read. Such correlations could be a function of the pre-training data: *e.g.*, perhaps politness varies with *formality* in the training data, both of which correlate with increased reading difficulty. However, a steerable model should be able to independently manipulate each textual aspect even given such dataset biases. In safety-critical settings, understanding and controlling side-effects is critical to evaluating the feasibility of LLM usage. Our findings motivate future study on the prevalence and (un)desirability of side effects in LLM behavior, and strategies for controlling undesirable side effects.

## 4 RELATED WORK

**Steerability in LLMs and generative models.** Many interventions for LLM steerability directly update model weights, such as activation steering (Turner et al., 2023; Rimsky et al., 2023; Konen et al., 2024), mono-semantic feature scaling (Templeton, 2024), or model-guided generation (Dathathri et al., 2020; Khalifa et al., 2021), which elicits pre-specified changes in text aspects during generation. Others leverage prompting, *e.g.*, instance-specific hints (Li et al., 2024b), global control codes (Keskar et al., 2019) and persona-based prompting (Li et al., 2024a; Liu et al., 2024a), which aim to align LLMs with pre-specified constraints or goals. While these methods aim to improve steerability, in contrast, we focus on first developing a principled framework for quantifying steerability. Sorensen et al. (2024) also propose a definition of "pluralistic steerable models," which is closest to our definition of steerability. We highlight that our steerability probe is one of the first practical instances of a "trade-off steerable benchmark" as proposed in their position paper.

We acknowledge that discourse on controllable generative models predates LLMs: the steerability of latent factors (Jahanian et al., 2020; Spingarn-Eliezer et al., 2021) and disentangled representation learning (Higgins et al., 2018; Locatello et al., 2019) are well-studied in generative adversarial networks, and more recently in large text+image models (Liu et al., 2022; Gavrikov et al., 2024) and graph generators (*i.e.*, molecule editing; Liu et al. (2024c); Du et al. (2022)). Latent factors are similar to our notion of *goals*, and framing disentangled representations Higgins et al. (2018) as independently manipulable subspaces (*i.e.*, *goals*) is reminiscent of our notion of side-effects. Our framework is potentially applicable to assessing the steerability of generative models beyond LLMs.

**Multi goal/objective-aware text generation.** Past works concerning LLM alignment towards multiple potentially-competing objectives or goals have proposed model-aggregation approaches (Rame et al., 2024; Jang et al., 2023) and RLHF variants with multi-goal rewards (Dong et al., 2023; Wang et al., 2024). The multi-goal objective function used by Wang et al. (2024) is closest to our definition of steerability, which also uses a vector-based model of user preferences based on attribute dimensions. However, rather than anchoring to the text generation process, we design a general probe for measuring and *comparing* out-of-the-box steerability in existing LLMs.

**Understanding and probing LLM in-context behavior.** Behavioral probes for LLMs include mechanistic interpretability approaches (Bricken et al., 2023; Templeton, 2024), which leverage sparse coding to extract interpretable dimensions of LLM behavior from model parameters. Other works explore trade-offs in LLM behavior, such as helpfulness vs. harmfulness (Liu et al., 2024b), or alignment with political biases in the United States (Liu et al., 2024a). Steerability can be seen as a case of pluralistic alignment in-context, for which we contribute an empirical probe.

## 5 CONCLUSION

Our empirical results suggest that current LLMs are not steerable. Although the results highlight ample room for improvement in steerability, our findings do not contradict that LLMs are powerful instruction-followers. Rather, our work highlights steerability as an independently important criterion for LLM evaluation. We highlight possible directions for future work. Our steerability probe can be applied to evaluate how other design choices in LLM development, such as aspects of LLM pre-training, RLHF, or changes in decoding parameters (such as the sampling temperature), affect steerability. Assessing existing steerability interventions via our probe could also contextualize progress in controllable LLMs. On the theoretical side, analyzing steerability could surface measurable prerequisites for steerability and how they intersect with existing approaches to pre-training/fine-tuning and alignment (*i.e.*, RLHF). Ultimately, our work lays a systematic foundation for quantitative steerability evaluation.

## ACKNOWLEDGEMENTS

We thank members of the MLD3 group at the University of Michigan and members of the Augmented Learning Reasoning Group at Microsoft Research for providing feedback on early versions of this work. Special thanks to Siddharth Suri, Jennifer Neville, and Wanqiao Xu for helpful conversations about the project motivation. T.C. and A.S. completed the majority of this work at Microsoft Research.

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

## A    EXPERIMENTAL IMPLEMENTATION DETAILS

### A.1    DATA PRE-PROCESSING

For all datasets, NLTK is used for all word/sentence counting capabilities. All source texts with over 2048 words are dropped. We further describe data extraction steps for each constituent dataset in our analysis:

- **CNN/Dailymail**: We take the validation split of version 3.0.0 and use the `article` column as the source text. We use the first 1220 examples in the dataset.

- **BookSum**: We take the validation split and use the `chapter` column as the source text and randomly sample 300 book chapters. To avoid duplicate source texts, we use only sources with a corresponding summary from Sparknotes (out of multiple possible summary providers). We chunk each text into a maximum of 30-sentence blocks, creating multiple source texts per book chapter.

- **Reddit TIFU**: We take the train split and sample 3000 posts (before dropping for length), using the `documents` column as the source text. We add periods to paragraph endings (word characters followed directly by newlines) to break up sentences without changing the semantics.

- **SCROLLS (SummScreenFD)**: We take the `summ_screen_fd` subset of SCROLLS, which features summaries of TV episodes based on scripts, using the `output` column as the source text.

The above process results in four subsets of each dataset, which are concatenated to create our initial seed set of texts.

A.2 PROMPT ENGINEERING

**Direct templated-based prompt design.**   Recall that, as per our framing, prompts are textual samples from a distribution conditional on $\Delta^{(i)}$. As a starting point, we construct simple template-based prompts to instruct the LLM to rewrite text to modify three goals, which we call the "direct prompting strategy." The text follows the template:

```
Please rewrite the following, but make it [GOAL 1],
[GOAL 2], and [GOAL 3].  Provide only the rewritten
text and do not explain your response.
```

The suffix following the goals (*i.e.*, "provide only the rewritten text...") is appended to minimize explanatory prefixes in the LLM response (*e.g.*, "Sure! Here's a rewritten version of your text:"), which may introduce noise in the goal evaluation.

To fill in `[GOAL #]`, we define "positive adjectival" and "negative adjectival" phrases for each goal as described in Table 1. The positive adjectival phrase is used for $\Delta^{(i)} > 0$, while the negative phrase is used for $\Delta^{(i)} < 0$. We further apply "modifiers" to each phrase depending on the *magnitude* of $\Delta^{(i)}$ as follows:

- If $|\Delta^{(i)}| \leq 0.2$: add "`slightly`" (*e.g.*, `more/less` → `slightly more/less`)
- If $0.2 < |\Delta^{(i)}| \leq 0.5$: keep as is
- If $|\Delta^{(i)}| \geq 0.5$: add "`much`" (*e.g.*, `more/less` → `much more/less`)

| Goal | Pos. Adj. Phrase | Neg. Adj. Phrase |
|---|---|---|
| Reading level | `harder to read` | `easier to read` |
| Politeness | `more polite` | `less polite` |
| Anger | `angrier` | `less angry` |
| Disgust | `sound more disgusted` | `sound less disgusted` |
| Fear | `more fearful-sounding` | `less fearful-sounding` |
| Joy | `happier` | `less happy` |
| Sadness | `sadder` | `less sad` |
| Surprise | `sound more surprised` | `sound less surprised` |
| Text diversity | `use more diverse language` | `use less diverse language` |
| Text length | `more verbose` | `more concise` |

Table 1: Initial "positive" and "negative adjectival" phrases for template-based prompting. Positive (negative) means that the prompt should request an *increase (decrease)* in the aspect of interest.

Lastly, we shuffle the ordering of the goals to mitigate recency and primacy biases in the prompt design. As an example, we show one randomly-generated template-based prompt:

```
Please rewrite the following, but make it sound more
surprised, much sadder, and much more rude.  Respond
with only the rewritten text and do not explain your
response.
```

The $\Delta^{(i)}$ values for the above prompt were `surprise`: 0.250, `sadness`: 0.645, `politeness`: -0.532, all other aspects: 0. As a final quality check, we visually inspect a sample of the LLM outputs for each model to ensure that the generations make meaningful attempts to follow the instructions (*i.e.*, the LLM attempts to rewrite the text and does not refuse to do so).

A.3 ADJUSTED PEARSON CROSS-CORRELATION

Here, we motivate and describe our adjusted cross-correlation metric. Let $z_i^*$ be the $i$th component of some target goal $\mathbf{z}^*$, and $\hat{\mathbf{z}}_j$ be the $j$th component of the vector of observed movement in goal-space. To investigate the impact of side-effects, or cases where requesting one goal leads to changes in another goal, we could consider measuring the cross-correlation between requested goals and observed goal-space movement, or $\rho(z_i^*, \hat{z}_j)$.

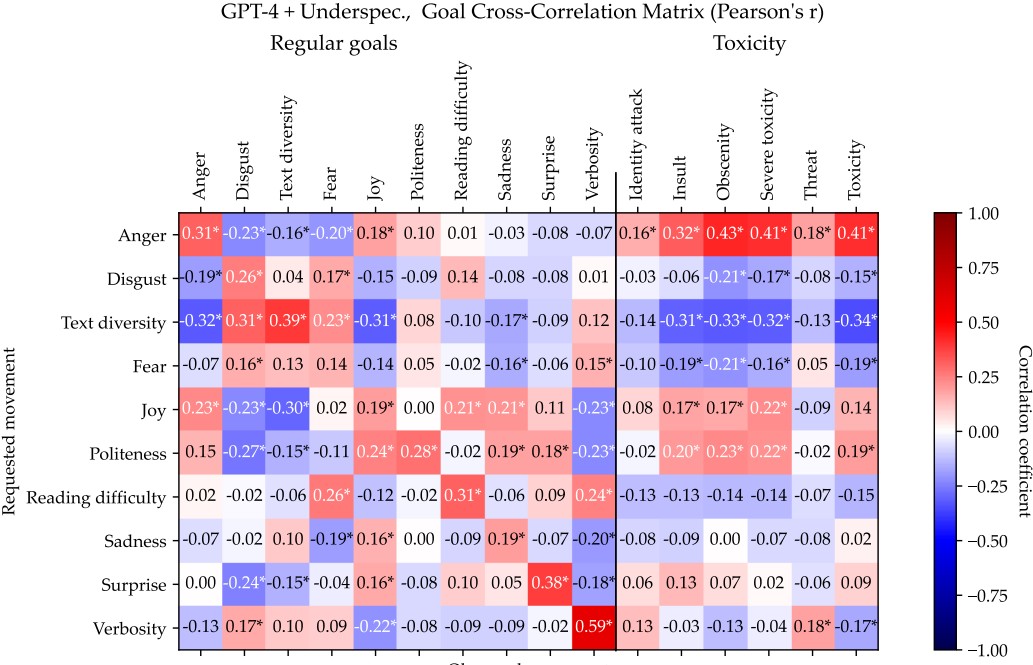

Figure 4: Naive cross-correlation matrix between requested goals and observed goal-space movement ($\rho(z_i^*, \hat{z}_j)$) before adjustment, vague/uninformative prompting strategy. "*" denotes statistical significance at $\alpha = 0.05$, with a Bonferroni correction for multiple hypotheses ($n = 160$).

However, without adjustment, there exists a "feasibility bias" in goal-space: the space of reachable target goals $\mathbf{z}^*$ is not invariant to $\mathbf{z}_0$, or the goal-space mapping of the source text.

**Why is an adjustment required?** Consider, for the sake of example, a simplified goal-space $[0, 1]^2$, and a source text with goal-space mapping $\mathbf{z}_0 = [0.01, 0.01]$. Note that this text lies near a "corner" of goal-space. Yet we know that $\mathbf{z}^*$ must lie in $[0, 1]^2$. Suppose that $\mathbf{z}^* = [0.5, 0.01]$, and make the simplifying assumption that a random rewrite is equivalent to randomly adding some noise to $\mathbf{z}^*$. Then a randomly-selected $\hat{\mathbf{z}}$ is more likely to increase the first dimension of $\mathbf{z}_0$ than the second dimension, which is desirable as per $\mathbf{z}^*$.

Thus, the presence of $\mathbf{z}_0$ in extremal regions of goal-space could potentially inflate the cross-correlation artificially. Hence, there is an implicit dependence between $\mathbf{z}_0$ and $\mathbf{z}^*$, while we are interested in capturing the dependence between $\mathbf{z}^*$ and $\hat{\mathbf{z}}$. When we measure $\rho(z_i^*, \hat{z}_j)$ naively, we risk capturing these two entangled effects.

**Computing adjusted cross-correlation using uninformative prompts.** The mechanism of correlation inflation is due to the dependence between $\mathbf{z}_0$ and $\mathbf{z}^*$. Thus, if we can measure the strength of this correlation, we could subtract it from naive measurements of $\rho(z_i^*, \hat{z}_j)$ to obtain an adjusted cross-correlation. To do so, we run a parallel steerability probe in which we replace our direct template-based prompt with a vague, uninformative prompt (*e.g.*, "Please rewrite this text with a few improvements."), leaving all other aspects of the steerability probe identical (*i.e.*, same source texts and user request vectors). The rewriting task with the uninformative prompt is intended to capture the inherent dependence between $\mathbf{z}_0$ and $\mathbf{z}^*$.

Formally, let $\rho_{vague}(z_i^*, \hat{z}_j)$ be the observed cross-correlation between the sampled $z_i^*$ and observed goal-space movement $\hat{z}_j$. Then, the adjusted cross-correlation $\tilde{\rho}(z_i^*, \hat{z}_j)$ is defined as

$$\tilde{\rho}(z_i^*, \hat{z}_j) \triangleq \rho(z_i^*, \hat{z}_j) - \rho_{vague}(z_i^*, \hat{z}_j). \tag{5}$$

Empirically, under an uninformative prompting strategy, we verify that cross-correlations may be artificially inflated (Figure 4), further justifying the adjustment.

Note that a potential alternative to adjusted cross-correlation is to filter out $\mathbf{z}_0$ that lie in extremal regions of goal-space before computing cross-correlation. However, due to the curse of dimensionality, a large proportion of texts would be filtered out under such a strategy: if we filter out points with any dimension less than $\varepsilon$ in $d$-dimensional space, we would only be able to keep $(1 - \varepsilon)^d$ points. For our steerability probe ($n = 2000$, $d = 10$), if $\varepsilon = 0.1$, only $\approx 34.9\%$ ($n \approx 697$) texts would be non-extremal in a perfectly uniform sample.

**Calculating statistical significance for differences in $\rho$.** To determine statistical significance for $\rho(\cdot)$, in general, Fisher's $z$-transformation is used as a variance-stabilizing transformation to ensure that the distribution of $\rho(\cdot)$ is asymptotically normal, with constant variance across values of $\rho$. In particular, for bivariate normal $z_i^*, \hat{z}_j$, $\rho$ is asymptotically distributed as $\mathcal{N}(\frac{1}{2}\ln(\frac{1+\rho}{1-\rho}), \frac{1}{N-3})$, where $N$ is the sample size used for computing $\rho$. Thus, $\tilde{\rho}$ has asymptotic variance $\frac{2}{N-3}$, and a standard normal CDF can be used to compute $p$-values for the resultant correlation coefficients.

## A.4 CHECKING GROUNDEDNESS

As another quality check on the generated outputs, we conduct a fuzzy grounded-ness check to ensure that the generated outputs stay reasonably on-topic with respect to the source text (*i.e.*, given "make the text angrier," the LLM does not simply produce unrelated angry sentences). A natural starting point is automated machine translation evaluation metrics such as the METEOR score (Banerjee & Lavie, 2005), since it accounts for both precision (parts of the generated text should overlap with the source text) and recall (parts of the source text should match the generated text). We base our metric on METEOR since it allows for fuzzy matching (full credit for using synonyms) and accounts for alignment between texts.

**METEOR-based groundedness evaluation.** For our purposes, we compute METEOR with *only* noun tokens (including proper nouns and pronouns), and weigh precision equally to recall. The restriction to nouns implicitly encodes our assumption that an ordered list of nouns is a sufficient statistic for the general topic and sequence of events in a text, while allowing for rewrites of the text to significantly change the underlying prose (*e.g.*, choice of adverbs and adjectives used).

Denote our modified meteor score as $\mathrm{M} : \mathcal{S} \times \mathcal{S} \to [0, 1]$. Intuitively, the distribution of METEOR scores between source text and their rewritten versions should differ significantly from the distribution of METEOR scores between source text and randomly-selected generated text (excluding rewritten versions). Thus, for each source text, we can compute the relative likelihood that each text was a rewritten version of the source text, rather than a randomly-selected example, using any probabilistic binary classification approach:

$$\text{Groundedness}(\mathbf{s}, \hat{\mathbf{s}}) = \frac{\hat{P}(\mathrm{M}(\hat{\mathbf{s}}, \mathbf{s}) \mid C = 1)}{\hat{P}(\mathrm{M}(\hat{\mathbf{s}}, \mathbf{s}) \mid C = 0)} = \frac{\hat{P}(C = 1 \mid \mathrm{M}(\hat{\mathbf{s}}, \mathbf{s}))}{\hat{P}(C = 0 \mid \mathrm{M}(\hat{\mathbf{s}}, \mathbf{s}))} \cdot \alpha \quad (6)$$

where $C = 1$ is the class of rewritten versions of $\mathbf{s}$, and $C = 0$ is the class of all other generated texts, and $\alpha = P(C = 0)/P(C = 1)$ is a normalization constant. Note that a groundedness value greater than one indicates that the generated text was more likely to have come from the distribution of rewritten text than the distribution of all other text, and vice versa.

All generated text that are more likely to be generated by the distribution of randomly-selected text than the distribution of rewritten text (as determined by statistical distance in METEOR score space) are manually reviewed.

**Limitations.** Note that this is not a hallucination detector. Rather, high values of groundedness mean that the generated outputs are likely to be on-topic, while low values mean that the generated outputs are potentially off-topic. Thus, our metric allows for hallucination. Second, this formulation *assumes* that we are using the LLM for a text rewriting task, such that there exists an expectation of groundedness/textual similarity pre- and post- LLM call.

**Implementation details.** We use Spacy's EN_CORE_WEB_SM pipeline as a part-of-speech tagger, entity recognizer, and tokenizer. We use NLTK's implementation of the METEOR score.

| Model | # of prompting failures ($n = 2000$) |
|---|---|
| GPT-3.5 | 1 |
| GPT-4-turbo | 0 |
| Llama3-8B | 4 |
| Llama3-70B | 0 |
| Mistral-7B | 0 |
| Mixtral-8x7B | 34 |

Table 2: Number of prompting/groundedness failures for each model (*e.g.*, refusals, truncated completions).

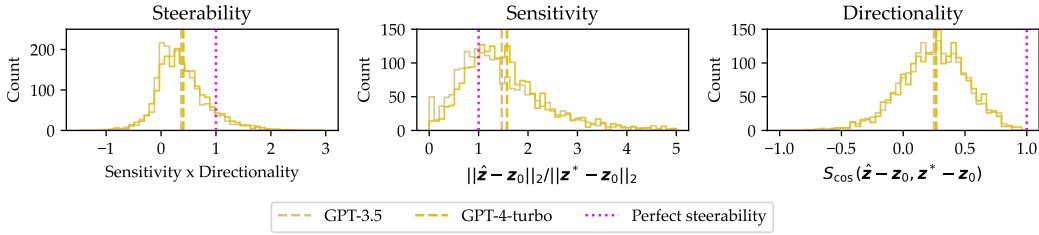

Figure 5: Sensitivity (left) and directionality (right) for GPT-3.5 (lighter) vs. GPT-4 turbo (darker).

**Statistics on prompting/groundedness failures.** We report the number of prompting/groundedness failures for each model evaluated in Table 2.

## B ADDITIONAL EMPIRICAL RESULTS

### B.1 STEERABILITY METRICS BY MODEL FAMILY

For clarity, we replot Figure 2, separated by model family:

- GPT family: Figure 5
- Mistral family: Figure 6
- Llama family: Figure 7

### B.2 ARE THE SELECTED PROMPTS TOO VAGUE?

As a secondary analysis of prompting, we evaluate our steerability metrics with respect to a more granular prompting strategy, in which models are instructed to change texts on a ten point scale (rather than "much/slightly/[no modifier]" + "more/less"). An example prompt (adapted from Appendix A.2) is

```
Please rewrite the following.  Assume that each
aspect of the text lies on a 10 point scale, where
1 represents the lowest possible level of that aspect,
while 10 represents the highest possible level.
Adjust the given aspects as follows:

- Increase the level of surprise of the text by 4
levels.
- Increase the level of sadness by 6 levels.
- Decrease the level of politeness by 5 levels.

Respond with only the rewritten text and do not
explain your response.
```

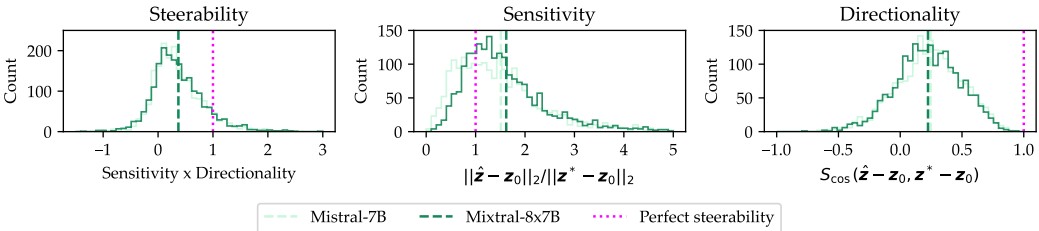

Figure 6: Sensitivity (left) and directionality (right) for Mistral (lighter) vs. Mixtral (darker).

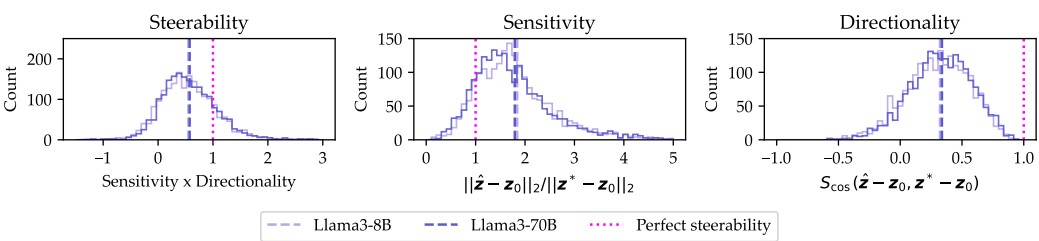

Figure 7: Sensitivity (left) and directionality (right) for Llama3-8B (lighter) vs. Llama3-70B (darker).

In summary, the more granular prompt has essentially no effect on directionality, but slightly decreases sensitivity for the Llama and GPT model families. Ultimately, the finding that LLMs tend to overshoot in goal-space still holds true.

For convenience, we provide links to the result figures here:

- GPT-4: Figure 8
- Llama3-70B: Figure 9
- Mixtral: Figure 10

### B.3 CROSS-CORRELATION BETWEEN GOALS FOR ALL MODELS

Here, we show side-effect plots (cross-correlation) for the other models in our evaluation, verifying that trends are similar. For convenience, we provide links here:

- GPT-3.5: Figure 11
- GPT-4: See body, Figure 3
- Llama3-8B: Figure 12
- Llama3-70B: Figure 13
- Mistral: Figure 14
- Mixtral: Figure 15

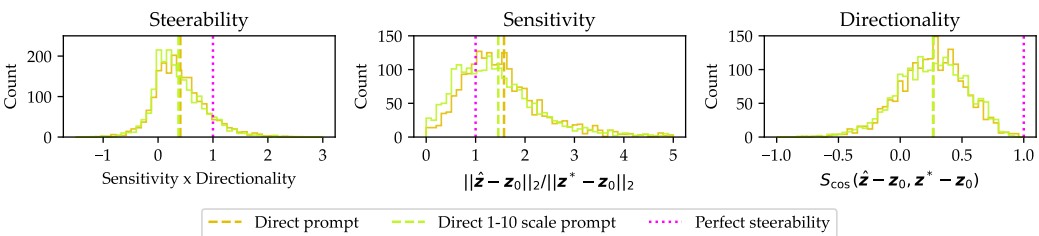

Figure 8: Sensitivity (left) and directionality (right) for GPT-4 turbo, comparing the template-based direct prompt (golden; Appendix A.2) and a more granular prompt based on a 1-10 scale (yellow-green).

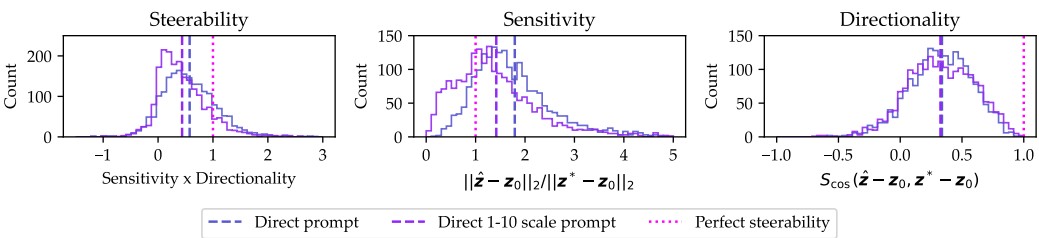

Figure 9: Sensitivity (left) and directionality (right) for Llama3-70B, comparing the template-based direct prompt (blue; Appendix A.2) and a more granular prompt based on a 1-10 scale (purple).

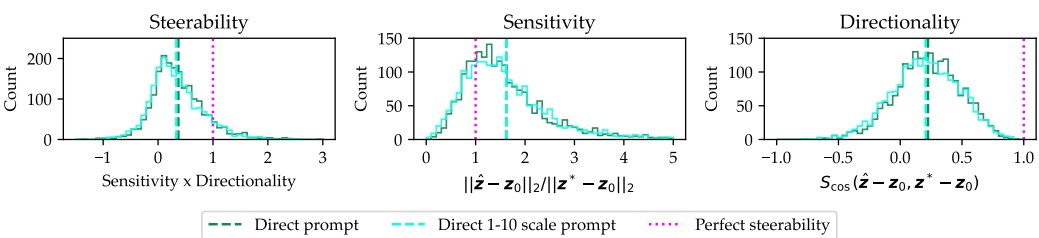

Figure 10: Sensitivity (left) and directionality (right) for Mixtral-8x7B, comparing the template-based direct prompt (green; Appendix A.2) and a more granular prompt based on a 1-10 scale (turquoise).

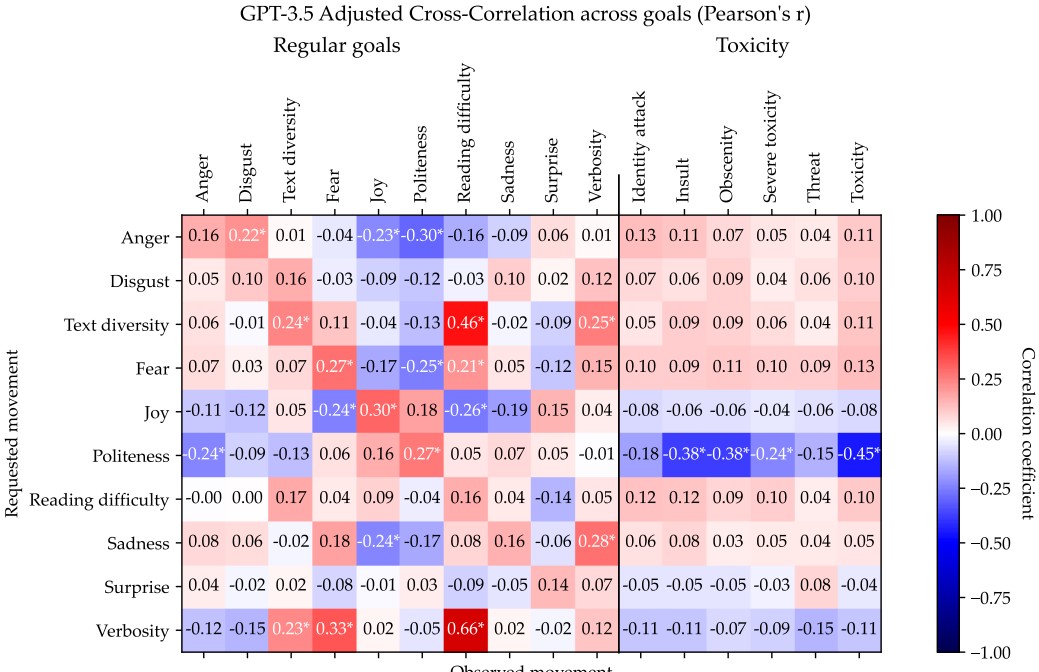

Figure 11: Adjusted cross-correlation matrix between requested goals and observed goal-space movement, GPT-3.5. "*" denotes statistical significance at $\alpha = 0.05$, with a Bonferroni correction for multiple hypotheses ($n = 160$).

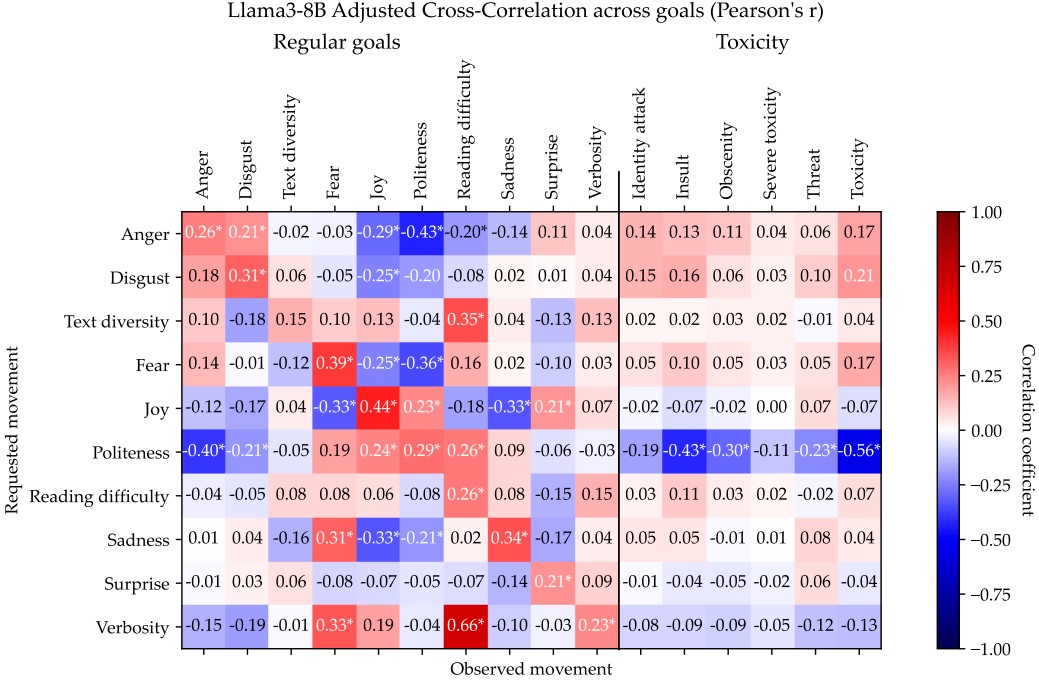

Figure 12: Adjusted cross-correlation matrix between requested goals and observed goal-space movement, Llama3-8B. "*" denotes statistical significance at $\alpha = 0.05$, with a Bonferroni correction for multiple hypotheses ($n = 160$).

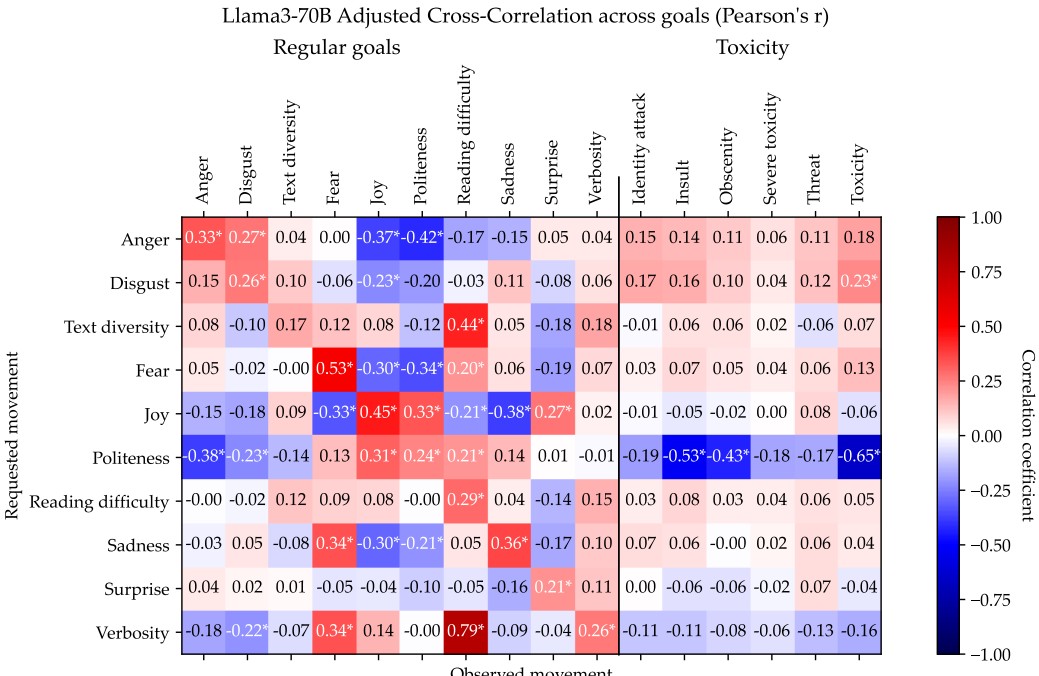

Figure 13: Adjusted cross-correlation matrix between requested goals and observed goal-space movement, Llama3-70B. "*" denotes statistical significance at $\alpha = 0.05$, with a Bonferroni correction for multiple hypotheses ($n = 160$).

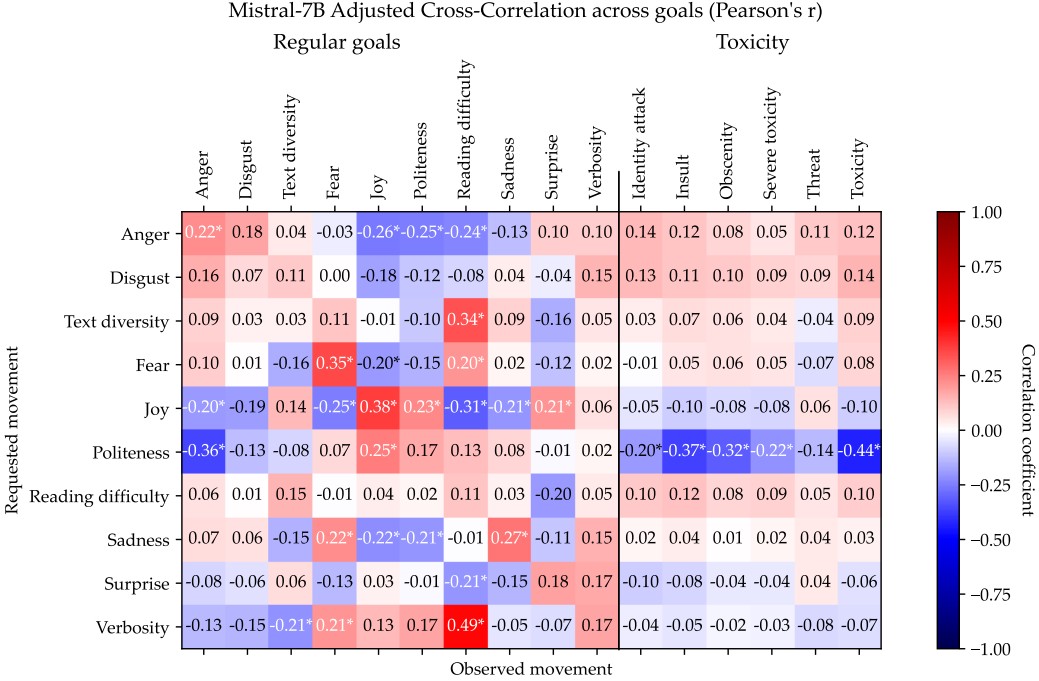

Figure 14: Adjusted cross-correlation matrix between requested goals and observed goal-space movement, Mistral-7B. "*" denotes statistical significance at $\alpha = 0.05$, with a Bonferroni correction for multiple hypotheses ($n = 160$).

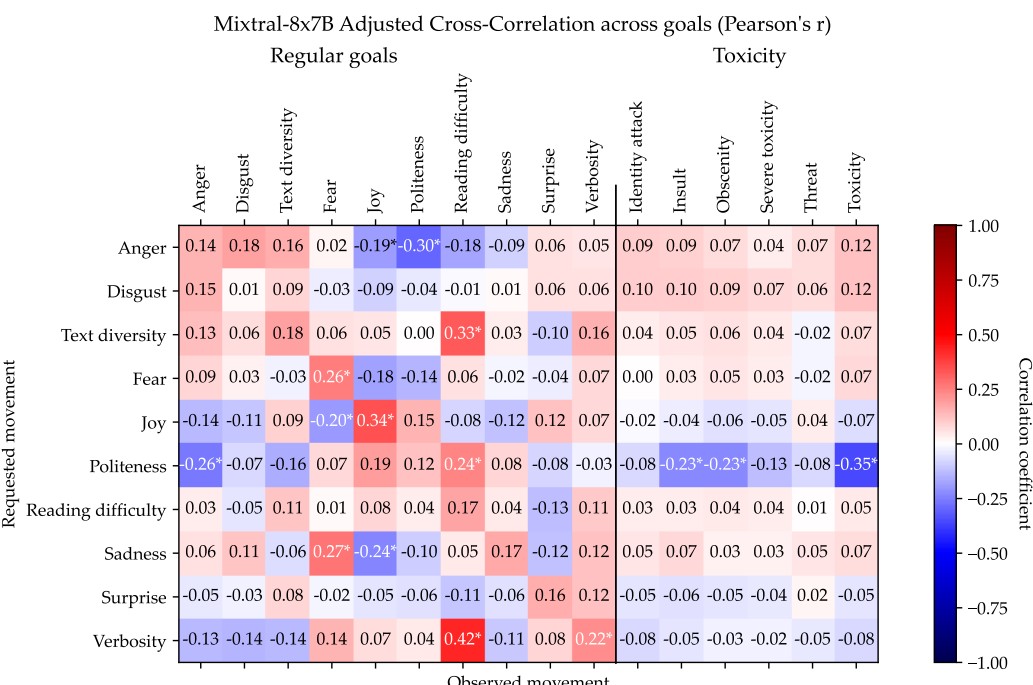

Figure 15: Adjusted cross-correlation matrix between requested goals and observed goal-space movement, Mistral-8x7B. "*" denotes statistical significance at $\alpha = 0.05$, with a Bonferroni correction for multiple hypotheses ($n = 160$).

