# OpenReview forum: "Measuring Steerability in Large Language Models"
_NeurIPS.cc/2024/Workshop/SafeGenAi — SafeGenAi Poster_

### Official Review · Reviewer_LqWg · 2024-10-09
**Measuring Steerability in Large Language Models**

**Rating:** 8
**Confidence:** 4

**Review:**

This paper centers on the concept of steerability in large language models (LLMs), highlighting the requirement for models that can be guided or “steered” to generate outputs that closely conform to specific user objectives. Although LLMs have demonstrated substantial enhancements in instruction-following capabilities, these advancements do not necessarily translate to improved steerability. The authors put forward a quantitative, goal-oriented definition of steerability, formulating it in terms of sensitivity (how responsive the model is to user prompts) and directionality (how well the model adheres to the desired direction of the goal).

The paper presents an empirical probe to measure steerability using text rewriting tasks, analyzing whether LLMs can adjust their outputs to reflect specific user-requested changes. It shows that even state-of-the-art LLMs such as GPT-3.5, GPT-4, Llama3, and Mistral have limitations in steerability, often deviating from user instructions and exhibiting unintended changes in non-requested directions. The study finds that larger models are not necessarily more steerable, attributing the lack of steerability to side effects where changes in one aspect inadvertently lead to shifts in other aspects.

1. The paper proposes a distinct and goal-oriented definition of steerability, which is both novel and practical.
2. This paper enables a robust evaluation of steerability across different LLM models, establishing a benchmark for future research.
3. The paper identifies and analyzes side effects when steering LLMs, such as unintentional changes in non-requested attributes.
4. By comparing models of different sizes and families (e.g., GPT, Llama3, Mistral), the study offers a nuanced understanding of how steerability varies across architectures.

---

### Official Review · Reviewer_Mgkv · 2024-10-09
**Interesting approach for added interpretability and steerability that could benefit from additional clarity and contextualization**

**Rating:** 4
**Confidence:** 3

**Review:**

Strengths

* The writing and paper are clean.  The authors present an interesting analyses on two-component ‘steerability’ with several language models
* The analyses includes directly measuring steerability as well as side effects.
* The “side effects” section was interesting in that it seems also generally relevant to working with LLMs, and shows that trying to alter outputs and strategies may have other unintended and perhaps unnoticed impacts


Weaknesses/Suggestions

* The biggest gap for me in the paper was understanding the methods.  The idea of looking at comparative vectors to probe steerability was clear, but I didn’t understand the use of the ‘goal space’ described the authors.  How are input and output texts represented in goal space?  Are the model tokens or weights involved?  This significantly impacts the utility/suitability of the method and was difficult to understand from the paper.
* How does the presented methodology compare to the other steerability methods mentioned in the Related Work section? What motivated defining this particular method?  This is touched on, but since other methods exist, it seems like it is important thing to clearly motivate this method
* Given that steerability is evaluated in a somewhat narrow scope (text rephrasing), additional context as to the importance of steerability in a practical sense would add to the introduction.
* It was hard to follow the “Defining Steerability” Section.  Describing the space z is in before describing probed directions in that space could have added helpful context.
* Why is a value of 1 necessary for steerability?  It seems like what value of steerability in the vector splace corresponded to clear steerability in the output space might be something to evaluate.
* It was difficult to follow the logic for the description of “overshooting” and large and small angles, though the content seems important — expanding on this description and/or including it in the Figure could improve clarity.
* The histograms of the steerability metrics overlap a lot, so it was hard to understand if/what was significant about the results
* The Conclusion states that the results suggest that LLMs are not steerable — is this supported by the literature?  If other methods have also shown poor steerability and this one does too, then this seems like a reasonable conclusion.  But if not, making this conclusion would require a more comprehensive evaluation of steerability than what is presented here.